# Adult Feeding Experience Determines the Fecundity and Preference of the *Henosepilachna vigintioctopunctata* (F.) (Coleoptera: Coccinellidae)

**DOI:** 10.3390/biology13040250

**Published:** 2024-04-09

**Authors:** Jingwei Qi, Xiangping Wang, Tingjia Zhang, Chuanren Li, Zailing Wang

**Affiliations:** Hubei Engineering Research Center for Pest Forewarning and Management, Institute of Entomology, College of Agriculture, Yangtze University, Jingzhou 434025, China; qijingwei77@163.com (J.Q.); wang.xiang.ping@126.com (X.W.); z15248891080@163.com (T.Z.); 13986706558@163.com (C.L.)

**Keywords:** *Henosepilachna vigintioctopunctata*, host plant, host switching, fecundity

## Abstract

**Simple Summary:**

Both larvae and adults of the *Henosepilachna vigintioctopunctata* (F.) (Coleoptera: Coccinellidae) can feed on potato, tomato, and eggplant leaves, though potatoes serve as the most suitable host for the *H. vigintioctopunctata*. Owing to the differing planting times of potatoes, tomatoes, and eggplants, *H. vigintioctopunctata* adults must migrate to tomato or eggplant leaves for feeding to ensure continuous food availability. Therefore, under wild field conditions, host transfer between larvae and adults of the *H. vigintioctopunctata* is a normal phenomenon. Generally, the feeding experiences of both larval and adult hosts influence the survival and reproduction of the adult ladybird beetle. To ascertain the impact of larval and adult hosts on the performance and preference of adults, we allow *H. vigintioctopunctata* larvae and adults to either continue or change their feeding experience on potato, tomato, and eggplant leaves. Our results indicate that the adult feeding host dictates the fecundity and preferences of the adult, independent of the larval feeding experience. While host switching between larval and adult stages yields fewer benefits for *H. vigintioctopunctata* performance compared to a consistent potato leaf diet, it facilitates food access for *H. vigintioctopunctata*. Hence, under wild field conditions, we can adjust the planting intervals between eggplants, tomatoes, and potatoes to prevent host switching between larva and adult *H. vigintioctopunctata*, thereby ecologically controlling the populations of *H. vigintioctopunctata*.

**Abstract:**

Both larvae and adults of the *Henosepilachna vigintioctopunctata* feed on leaves of potatoes, tomatoes, and eggplants. Given the variation in planting times of host plants in the Jianghan Plain, host switching between larvae and adults of *H. vigintioctopunctata* is inevitable to ensure continuous food availability. We evaluated the effect of consistent versus diverse larval and adult host plant feeding experience on growth performance, fecundity, longevity, and feeding preferences of *H. vigintioctopunctata* through match-mismatch experiments. Host plant quality significantly influences larval development and adult reproduction. Potatoes are identified as the optimal host plant for *H. vigintioctopunctata*, whereas eggplants significantly negatively affect the adult fecundity. Adult stage host feeding experience determines the fecundity of *H. vigintioctopunctata*, irrespective of the larval feeding experience. The fecundity of *H. vigintioctopunctata* adults on eggplant leaves remains significantly lower than that observed on potato leaves. Similarly, adult *H. vigintioctopunctata* demonstrate a preference for consuming potato leaves, irrespective of the larval feeding experience. Although host switching between larval and adult stages offers lesser benefits for the performance of herbivorous insects compared to a consistent diet with potato leaves, it maintains *H. vigintioctopunctata* population continuity amidst shortages of high-quality potato hosts.

## 1. Introduction

The nutritional content (proteins, carbohydrates, fats) and secondary metabolites of the host plant determine the growth, development, and reproduction of herbivorous insects [1,2], as well as the insects’ immunity and tolerance to biotic (natural enemies, pathogenic bacteria) [3,4,5] and abiotic factors [6,7]. However, the nutritional content and secondary metabolites of host plants is subject to change due to variations in plant varieties [5], growth phases [8] and environmental condition [9]. Consequently, herbivorous insects reared on plants of differing quality exhibit significant variations in development time and life history traits, such as fecundity, and hatchability [10,11,12]. To mitigate the effects of environmental heterogeneity on insect growth, development, and reproduction, certain polyphagous insects strategically optimize nutrient intake by consuming mixed diets or alternating between host plants, thus minimizing adaptive damage [13]. For instance, larvae of *Grammia incorrupta* (Edwards, 1881) (Lepidoptera, Arctiidae) consume various host plants throughout their development to aids in meeting nutritional requirements and circumventing excessive intake of defense compounds, thereby enhancing immunity and resistance to pathogens and parasites [14,15,16].

However, owing to the restricted mobility of most herbivorous insect larvae, particularly Diptera and Coleoptera, these larvae are incapable of moving among multiple hosts [17]. Given that host plant availability and quality vary seasonally, adults may encounter hosts different from those experienced by their larvae [18]. Therefore, host switching from larvae to adults is more prevalent among herbivorous insects, particularly when both life stages feed on the plant [19,20]. However, the decision-making process of herbivorous insects regarding the feeding on a variety of food sources and the switching of hosts, either between or within host plants, is influenced by the costs and benefits associated with such behavior [13]. For instance, the behavioral and physiological adjustments of herbivorous insects to their original host can influence their successful adaptation to and utilization of new hosts, which, in turn, impacts their development and reproductive success [21,22,23]. At the same time, herbivorous insects are constrained by biological factors such as predation risk, competition, or parasitism during host switching, thereby reducing the possibility of host switching between larvae and adults [24]. Generally, the advantages of host switching between larvae and adults surpass the associated costs. This necessity arises as most adults require food post-emergence to fulfill their nutritional needs essential for growth and reproduction, even in the face of external environmental interference.

Host selection and host switching can occur concurrently during the feeding phase of newly emerged adults [25]. The selection of diverse host plants by new emerged adults are a critical determinant in the host-switching process between larvae and adults. However, the purported strong correlation between larval experience and the host plant preferences of eclosion adults remains contentious [26]. For example, Silva (2014) [27] highlight that feeding experiences during the larval stage may influence adult host selection, resulting in a tendency for adults to favor familiar hosts over other acceptable or potentially superior host plants. Additionally, Bernays (1996) [28] and Del Campo (2000) [29] also note that adults exclusively accept familiar hosts as suitable for colonization. However, Cai (2014) [30] and Xiao (2023) [31] argue that certain volatile compounds, such as (Z)-3-hexenyl acetate and monoterpenoids, produced by plants, can influence adults’ selection of new host plants. Consequently, emerging adults often exhibit an “innate” preference for certain hosts, despite lacking prior experience with these hosts during the larval stage. Alternatively, herbivores may preferentially feed on hosts whose nutritional content is more beneficial to insect development [32,33,34]. Therefore, examining the impacts of host choice and host switching on the life history performance (lifespan and reproduction) of adult stages offers a comprehensive understanding of the ecological and evolutionary determinants of these behaviors [35]. Match-mismatch experimental designs maintaining the experimental population in identical environmental conditions or transferring it to varied environmental conditions after a specified period are frequently utilized to study host selection and host-switching between larvae and adults [1,36,37]. These designs facilitate the investigation of the significance of specific life stages in influencing insect life history performance and preferences, offering insights into the underlying mechanisms and functions [35].

*Henosepilachna vigintioctopunctata* (Fabricius) (Coleoptera: Coccinellidae), also known as the 28-spotted potato lady beetle in China, is an oligophagous insect and one of the most economically significant pests of eggplant, tomato, potato, and other Solanaceae plants in the Jianghan Plain [38,39]. Both adults and larvae of *H. vigintioctopunctata* feed on the leaves of eggplant, tomato, and potato. The planting period for eggplant and tomato spans from April to November, whereas the planting time for potatoes typically extends from November of the previous year to May of the following year in the Jianghan Plain. Although the planting time differences between eggplant, tomato, and potato prompt a host switch between the larvae and adults of *H. vigintioctopunctata*, they also ensure a continuous food source for this species. Therefore, in this study, we utilized the match-mismatch experiment to (1) investigate the effects of constant versus changing larval and adult feeding experiences on the life history of *H. vigintioctopunctata*; (2) explore the correlation between larval stage host food and adult stage selection preferences; and (3) analyze the correlation between the performance and preference of the *H. vigintioctopunctata* for different hosts and the nutritional content of the host plants.

## 2. Materials and Methods

### 2.1. Host Plant

The major cultivars grown throughout the Jianghan Plain include the potato *Solanum tuberosum*, tomato *Solanum lycopersicum* (L402, Xi’an Hejia Seed Co., Ltd., Xi’an, China), and eggplant *Solanum melongena* (Eyou, Wuhan Hongda Seed Co., Ltd., Wuhan, China), which were cultivated in the greenhouse at 25 ± 2 °C under a 16:8 h light-dark regime. Plants were grown in plastic pots (20 cm in diameter and 16 cm in height, filled with sandy loam) without the use of pesticides in the same greenhouse.

### 2.2. Insect Material

Ten pairs of *H. vigintioctopunctata* adults were collected from host plants of potato in Jingzhou city, China (112°18′ S, 30°35′ E). A stock colony of *H. vigintioctopunctata* was reared on potato and maintained in Petri dishes (140 mm in diameter) in an incubator (GZX-400BS-III, Shanghai Xin-Miao Medical Equipment Manufacturing Co., Ltd., Shanghai, China), set at 26 ± 1 °C, with a 16:8 (L:D) photoperiod and 75 ± 10% humidity. All of the eggs used in this study were collected from second-generation, laboratory-reared *H. vigintioctopunctata* adults.

### 2.3. Match-Mismatch Experimental Design

To investigate the impact of consistent and variable host plant availability on the life history performance of *H. vigintioctopunctata*, nine different match-mismatch experimental designs were established. We selected 900 larvae that hatched on the same day to constitute our experimental population. Subsequently, we divided these 900 larvae into three equal groups, with each group consisting of 300 larvae. Each group was then raised on a different host of leaf: potato, eggplant, or tomato, until all larvae reached pupation. We placed 10 *H. vigintioctopunctata* larvae in petri dishes with filter paper and established 30 replicates to compare the effects of potato leaves, tomato leaves, and eggplant leaves on the development and survival of *H. vigintioctopunctata* larvae. The development period and survival rate of *H. vigintioctopunctata* larvae on eggplant, potato, and tomato were recorded daily, and the weight of 30 *H. vigintioctopunctata* pupae on different hosts was measured using an Ohaus balance (Model AX2242H/E, Ohaus, Parsippany, NJ, USA). Subsequently, we divided the emerging adults that larvae fed on the same host plants into three equal parts for match-mismatch experiments. *H. vigintioctopunctata* larvae were initially fed potato leaves, with one-third of the newly emerged adults subsequently fed potato leaves (Potato-Potato, PP), while the remaining two groups were fed eggplant leaves (Potato-Eggplant, PE) and tomato leaves (Potato-Tomato, PT), respectively. Similarly, *H. vigintioctopunctata* larvae were initially fed on eggplant leaves, and then one-third of the newly emerged adults were fed eggplant (Eggplant-Eggplant, EE), potato (Eggplant-Potato, EP), or tomato (Eggplant-Tomato, ET) leaves, respectively. In a similar vein, *H. vigintioctopunctata* larvae fed on tomato leaves, and then one-third of the newly emerged adults were fed tomato (Tomato-Tomato, TT), potato (Tomato-Potato, TP), or eggplant (Tomato-Eggplant, TE) leaves, respectively (Figure 1). The preoviposition period, oviposition period, fecundity, and the longevity of both female and male adults across these nine treatments was documented until all adults died. At the same time, the preadults survival rate of *H. vigintioctopunctata* on eggplant, tomato and potato dictated the number of replicates in the match-mismatch experiments. We adopted the two-sex life table to solve the condition that the male-to-female ratio during the experiment was not 1:1 [40]. When the number of females exceeded that of males, we arranged for mating between the surplus females and males not involved in the experiment, subsequently tracking the fecundity and longevity of *H. vigintioctopunctata* female. Conversely, when males outnumbered females, we maintained the males separately, monitoring their longevity under each treatment until death. The numbers of female and male replicates across different treatments are shown in Appendix A.

### 2.4. Adult Feeding Preference Experience

Newly hatched *H. vigintioctopunctata* larvae were fed on potato, tomato, or eggplant leaves until pupation. For each host plant, 50 *H. vigintioctopunctata* female that emerged on the same day were selected for feeding preference experiments. One potato, tomato, and eggplant plant were placed in three corners of a net chamber (40 cm high, 90 cm long, 50 cm wide). We then placed ten newly emerged *H. vigintioctopunctata* females, previously fed on the same host plant (potato, tomato, or eggplant), in the center of the net chamber to ensure that they were equidistant from the three host plants, allowing free choice. After 24 and 48 h, the number of *H. vigintioctopunctata* adults on each plant was recorded. Simultaneously, after 48 h, all potato, tomato and eggplant leaves damaged by the *H. vigintioctopunctata* adults were removed from the net chamber, scanned with a flatbed scanner, and the consumed area was estimated using ImageJ (https://ij.imjoy.io/, accessed on 3 May 2023). The consumed mass by *H. vigintioctopunctata* adults on different host plants was calculated based on the mass of known areas of each plant. We established the linear correlation between the weight and area of host plant leaves. Subsequently, we calculated the feeding weight of *H. vigintioctopunctata* on various host plants based on the feeding area. The preference experiment for *H. vigintioctopunctata* adults in each host plant treatment was conducted in five replicates.

### 2.5. Nutrient Components of Host Plant Leaves

The water content, crude fat, total protein, total carbohydrate and total amino acid contents in young potato, tomato and eggplant leaves were measured. (1) Water content: 0.1 g of fresh leaves was placed into a 1.5 mL centrifuge tube. The total weight (W1) of the leaves and centrifuge tube was measured, and then the centrifuge tube containing the fresh leaves was placed in an oven set at 60 °C for 24 h, followed by cooling for 2 h, before weighing the centrifuge tube and leaves again (W2). Water content = W1 − W2; (2) crude fat: 2 mL of a chloroform and methanol mixture (chloroform: methanol = 2:1, *v*/*v*) was added to the centrifuge tube containing the dry leaves, and then the dry leaves were ground into a homogenate. We removed the supernatant after the centrifuge tube was centrifuged at 12,000 rpm for 10 min. Subsequently, we added 2 mL of the mixture to the centrifuge tube and repeated the centrifugation once. The centrifuge tube residue (W3) was weighed after being placed in an oven at 60 °C for 72 h. Crude fat content is calculated as W2 − W3. (3) Total protein, carbohydrate and total amino acid content: A plant protein quantification kit (Coomassie Brilliant Blue method), total amino acid determination kit, and soluble sugar content test kit (colorimetric method) (Nanjing Jiancheng Bioengineering Institute) were utilized to measure the total protein, total soluble sugar, and total amino acid content in host plant leaves, respectively.

### 2.6. Data Analyses

Univariate ANOVAs were conducted to assess the impact of the host plant on larval and pupal development times, survival rates, and the pupal dimensions (weight, length, and width) of *H. vigintioctopunctata*. Similarly, in the feeding preference experiment, ANOVAs were conducted to assess the weight of consumed by *H. vigintioctopunctata* on different host plants. Generalized linear mixed-effects models (GLMMs) were utilized to analyze the effects of larval and adult feeding experiences on the performance and preference of adult ladybirds. For the statistical outputs analyzed with GLMMs, tables of variance analysis displaying *F*- and *p*-values are presented. Post-hoc analyses based on the GLMMs were employed to compare significant differences in the preoviposition and oviposition periods, as well as the longevity (female and male) and fecundity of *H. vigintioctopunctata* adults, across nine treatments.

## 3. Results

### 3.1. Effects of Host Plants on the Development of Larvae and Pupae of H. vigintioctopunctata

No significant difference was observed in the mean development times (d) of the first instar (L1) and pupa, as well as in the pupal length (mm) of *H. vigintioctopunctata* reared on potato, tomato, and eggplant. The development period of the second instar (L2) of *H. vigintioctopunctata* reared on tomato (2.62 d) was significantly shorter than that on potato (2.88 d) and eggplant (3.17 d). No significant difference was found in the development period of the third instar (L3) of *H. vigintioctopunctata* reared on tomatoes (2.82 d) and potatoes (2.85 d), yet this period was significantly shorter than on eggplant (3.75 d). The development time of the fourth instar (L4) of H. vigintioctopunctata fed on potato leaves (5.18 d) was significantly shorter than that on tomatoes (5.63 d) and eggplant (6.0 d). Although the total larval stage of *H. vigintioctopunctata* reared on potato (14.28 d) was significantly shorter than on tomato (15.11 d) and eggplant (16.61 d), the larval survival rate reared on potato (68%) was significantly lower than on tomato (91%) (Table 1). The pupal weight and width of *H. vigintioctopunctata* reared on eggplant (15.62 mg and 6.69 mm) was significantly lower than those reared on potato (21.14 mg and 6.89 mm) (Figure 2).

### 3.2. Effect of the Larval and Adult Host Plant Experience and Their Interaction on H. vigintioctopunctata Adult Performance

Larval host feeding experience did not significantly influence the performance of *H. vigintioctopunctata* adult. Adult host experience and the interaction of larva * adult host experience did not significantly affect the preoviposition period, female and male longevity of *H. vigintioctopunctata* adult. The adult host experience significantly affects the oviposition period and fecundity of *H. vigintioctopunctata* adult. The interaction of larva * adult host experience significantly affects the fecundity of *H. vigintioctopunctata* adult (Table 2). The oviposition period of the adult on treatment of EE (both larval are adult host experience are eggplant) is significantly shorter than on EP (88.75 d), PP (76.68 d), ET (76.29 d) and PT (89.20 d) (Figure 3A). The fecundity of the *H. vigintioctopunctata* adults on treatment of EE (85.83 egg/female), TE (114.46 egg/female) and PE (102.21 egg/female) were significantly smaller than that on EP (441.84 egg/female) and PP (625.57 egg/female), TP (596.63 egg/female) and ET (478.71 egg/female) (Figure 3B). This finding indicates that regardless of whether the larvae consume potato, tomato, or eggplant leaves, if the adults consume eggplant leaves, their reproduction will be significantly lower compared to when they feed on potato leaves.

### 3.3. Effects of Larval Host Experience on Adult Preference and Feeding of H. vigintioctopunctata

In the selective experiment, the feeding preference of *H. vigintioctopunctata* adults for potatoes, tomatoes, and eggplants was not influenced by larval host experience, but was primarily determined by the adults’ preference for host plants (Figure 4). In our experiment, irrespective of whether the larvae consumed potato leaves, tomato leaves, or eggplant leaves, the selection rate of newly emerged adults for potato leaves (60–74%) was higher than that for tomatoes (6–10%) and eggplant (6–16%) leaves (Figure 4). Additionally, *H. vigintioctopunctata* adults make a choice regarding the host plant within 24 h, with no significant difference in host preference observed at 48 h. The amount of food consumed by *H. vigintioctopunctata* adults on potatoes (3.5 g) is 2.8- and 1.8-fold greater than that on eggplants (1.25 g) and tomatoes (1.9 g), respectively (Figure 5).

### 3.4. The Water and Nutritional Content of Host Plants

The water and nutrient contents in potato, tomato, and eggplant leaves are depicted in Figure 6. The water content of potato leaves (848 mg/g) was the highest, significantly higher than that of tomato leaves (782 mg/g) and eggplant leaves (700 mg/g). The crude fat content in tomato leaves (49 mg/g) was significantly higher than in eggplant leaves (30 mg/g) and potato leaves (31 mg/g). Proteins and carbohydrates are the most important nutrients found in plant leaves. The protein content in potato leaves (48 mg/g) is significantly lower than in tomato leaves (67 mg/g) and eggplant leaves (68 mg/g). The carbohydrate content in tomato leaves was significantly higher than in potato leaves and eggplant leaves. The total amino acid content in potato leaves (1.0 mg/g) was significantly higher than in tomato leaves (0.78 mg/g), but not significantly different from eggplant leaves (0.88 mg/g) (Figure 6).

## 4. Discussion

### 4.1. Effects of Host Plant Nutritional Differences on the Performace H. vigintioctopunctata

Plant nutrients are crucial factors affecting insect growth, development, reproduction, and population dynamics [1]. Larvae require nutrients for growth and to progress to the next stage [41], while adults need them for ovary development and fecundity [42,43]. Therefore, for herbivorous insects, especially both larvae and adults consume the same plants, the identical diet may not satisfy the distinct nutritional needs of *H. vigintioctopunctata* larvae and adults. For instance, in 2018, our research team successfully raised *H. vigintioctopunctata* from larvae to adults using pure artificial feed (without any plant ingredients), achieving a larval survival rate as high as 72%, which was not significantly different from that observed with the natural host plant. However, this artificial feed can sustain the survival of *H. vigintioctopunctata* adult but fails to fulfill the nutritional requirements for fecundity [44]. Similarly, Kawazu (2014) [45] reported that larvae could be raised on artificial feed, but adults required plant leaves to meet their nutritional needs for reproduction. In our experimental results, the development period of *H. vigintioctopunctata* larvae on eggplant leaves was significantly longer by 14.1% (2.33 d) compared to those continuously fed on potato leaves, while the fecundity of adults on eggplants dropped significantly by 86.3% (539.3 eggs/female). Therefore, compared with the nutrients required by *H. vigintioctopunctata* larvae, the adult beetles have more stringent requirements on the nutritional content of host plants.

The performance in terms of larval development period and adult fecundity of *H. vigintioctopunctata* on potato leaves surpasses that observed on tomato and eggplant leaves, indicating that potatoes serve as the most suitable hosts for *H. vigintioctopunctata*. A higher water content and an optimal ratio of protein to carbohydrates may contribute to the superior performance of larvae and adults of *H. vigintioctopunctata* on potato leaves. For phytophagous insects, leaves of plants with high water content tend to be more tender and easier to consume, facilitating easier digestion and absorption, thereby promoting rapid growth of the insects [46]. Furthermore, despite the lower protein and carbohydrate content in potato leaves compared to tomatoes, an optimal ratio of protein to carbohydrates often proves beneficial for the growth, development, and reproduction of herbivorous insects [47,48]. Numerous researchers utilizing the Geometric Framework for Nutrition (GFN) model have demonstrated that an optimal ratio of protein to carbohydrates enables insects to achieve superior growth, development, and reproductive performance. High protein and low carbohydrate diets, or conversely, low protein and high carbohydrate diets, adversely affect the growth, development, and reproduction of insects [49,50,51,52]. For example, Lee (2008) [51] and Fanson (2009) [53] found that when *Drosophila melanogaster* and *Bactrocera tryoni* were fed diets with a lower protein-to-carbohydrate ratio (p:c = 1:16), they exhibited the longest adult lifespans but the lowest fecundity. With an increasing protein proportion in their diet, adult longevity decreases while fecundity increases. At a protein-to-carbohydrate ratio of 1:4, the fecundity of *Drosophila melanogaster* and *Bactrocera tryoni* peaks. However, to date, no researchers have employed the GFN model to confirm whether the optimal protein and carbohydrate ratio for *H. vigintioctopunctata* adults closely matches that found in potato leaves. Although our research group has determined the most suitable protein and carbohydrate ratio for the larvae of *H. vigintioctopunctata* through the GFN model, it has proven impossible to ascertain the most suitable protein and carbohydrate ratio for *H. vigintioctopunctata* adults due to the ineffectiveness of pure artificial feed [44,54]. This limitation underscores the need for further research and breakthroughs.

Indeed, the secondary metabolites in eggplant leaves may also contribute to the decline in fecundity of *H. vigintioctopunctata* females. Glycoalkaloids, which are predominant among the plant secondary metabolites in eggplant, tomato, and potato, aid in these plants’ resistance to insect and pathogen damage [55]. Among different Solanaceae species, the types and concentrations of glycoalkaloids significantly vary. For example, tomato plants primarily contain tomatine and dehydrotomatine as glycoalkaloids, while α-solanine and α-chaconine are predominant in potato and eggplant leaves [56]. In eggplant leaves, 95% of the glycoalkaloids consist of α-solanine [57]. The α-solanine content in eggplant leaves (600 µg/g) exceeds that in potato leaves (83.5 µg/g) [58,59]. Various research studies have demonstrated that α-solanine significantly decreases the survival rates of larvae and pupae, as well as adult fecundity [60,61]. Simultaneously, Devanand (2011) [62] extracted a glycoalkaloid mixture from eggplant that significantly interferes with insect molting, thereby reducing larval survival rates. Additionally, eggplant leaves are enriched with a diverse array of secondary metabolites, including caffeoylquinic acid derivatives, flavonoids, and saponins [63], which also aid eggplants in defending against damage by herbivores. For instance, when eggplant and tomato are subjected to *Tuta absoluta* infestations, eggplant leaves produce a significant quantity of volatile organic compounds and terpene-based primary/secondary metabolites to defend against *Tuta absoluta* infestations, in contrast to tomatoes, which produce minimal or none [64]. Concurrently, our metabolomic and transcriptomic analyses indicated that secondary metabolites in eggplant leaves cause a reduction in ecdysone levels within the steroid metabolism pathway in the hemolymph, resulting in decreased expression of downstream vitellogenin genes, which in turn impairs ovarian development and ultimately reduces fecundity in *H. vigintioctopunctata* (unpublishing).

### 4.2. Effect of Larval and Adult Feeding Experience on the Fecundity of H. vigintioctopunctata Adult

The silver spoon effect suggests that the nutrients obtained by larvae can confer lasting growth and reproductive advantages to adults. In other words, the quality of larval nutrition directly influences the life-history traits of adults [65]. Generally, when larvae are provided with optimal nutritional conditions, the resulting adults tend to be larger and more competitive [66,67]. In our experiments, the pupal weight of *H. vigintioctopunctata* on potato was significantly higher than that on eggplant leaves (Table 1), indicating that adults emerging from larvae that fed on potato leaves are larger than those emerging from larvae that fed on eggplant leaves. However, even with larvae that have experienced feeding on potatoes, the fecundity of adults subsequently fed on eggplant leaves (PE) remains significantly lower than that of adults on potato leaves and whose larvae stage involved feeding on eggplant leaves (EP) (Figure 3). In contrast, the environmental matching hypothesis suggests that this silver spoon effect depends on continued favorable conditions during the adult stage [65]. Due to compensatory abilities in the adult stage, the host plant during the larval stage does not significantly impact adult fecundity and survival, which suggests that host plant in the adult stage plays a decisive role in the adaptability of the adult [68,69]. Therefore, for *H. vigintioctopunctata*, the environmental matching hypothesis more aptly explains the variation in larval and adult feeding experiences and their influence on adult reproductive performance. Simultaneously, by analyzing egg or embryo initiation and maturation processes, we can effectively elucidate the distinctions between the silver spoon effect and the environmental matching hypothesis. In Lepidoptera, all eggs are present in the ovarioles upon adult eclosion [70]. Thus, the nutrient levels received by larvae dictate the adult’s fecundity [19]. However, in Coleoptera [17] and aphids [71], egg or embryo initiation and maturation can persist throughout the adult’s reproductive lifespan. Therefore, nutrients acquired during the adult stage are crucial in determining achieved fecundity.

### 4.3. The Advantage of Host Switch between Larva and Adult for H. vigintioctopunctata

Host switching between larval and adult stages offers lesser benefits for the performance of herbivorous insects compared to a consistent diet with an optimal food, particularly when the alternative host exhibits potent chemical defenses against herbivores [16]. Consequently, it is recommended that both larval and adult stages of *H. vigintioctopunctata* consume potato leaves to optimize population expansion in ideal conditions. However, the varied planting time of potatoes, tomatoes, and eggplants in the Jianghan Plain necessitate host switching between the larval and adult stages of *H. vigintioctopunctata* to ensure a continuous food supply. As previously mentioned in introduction, host switching between larvae and adults involves a trade-off between benefits and drawbacks. Early field investigations by our research team in the Jianghan Plain revealed that following potato harvests, the presence of eggs and larvae of *H. vigintioctopunctata* on eggplant leaves surged, which means that the scarcity of potato leaves prompts the *H. vigintioctopunctata* to lay eggs on tomato or eggplant leaves [39]. Our experiments demonstrate that, compared to scenarios where both larvae and adults consume potato leaves, adult fecundity declines by 83.7% when larvae are fed potatoes and adults are subsequently fed eggplants. However, after larvae consumed eggplant leaves, adults feeding on potatoes and tomatoes exhibited a significant increase in egg-laying, by 518% and 562%, respectively. This resulted in an increase in the population size of the *H. vigintioctopunctata*. Therefore, host switching between larvae and adults not only helps the ladybug adeptly navigate the challenges of losing its preferred host but also maintains population density.

## 5. Conclusions

The variation in planting times for potatoes, tomatoes, and eggplants in the Jianghan Plain results in inevitable host shifts between the larvae and adults of *H. vigintioctopunctata*. Through match-mismatch experiments, it was found that the fecundity and preference of *H. vigintioctopunctata* adults are not influenced by larval feeding experiences. In other words, the adult stage host plant determines the fecundity and feeding preference of *H. vigintioctopunctata* adults. Variations in water content and nutritional components (proteins, carbohydrates, crude fat, and total amino acids) in the leaves of potatoes, tomatoes, and eggplants result in significant differences in the development time, survival rate, and fecundity of *H. vigintioctopunctata*. Regardless of the larvae’s host plant feeding experience, adults feeding on eggplant leaves resulted in a significant decrease in fecundity. Similarly, regardless of the larvae’s host plant, adults displayed a preference for potato leaves. Although host switching between larval and adult stages offers lesser benefits for the performance of herbivorous insects compared to a consistent diet with potato leaves, it maintains *H. vigintioctopunctata* population continuity amidst shortages of high-quality potato hosts. Hence, under wild field conditions, we can adjust the planting intervals between eggplants, tomatoes, and potatoes to prevent host switching between larva and adult *H. vigintioctopunctata*, thereby ecologically controlling the populations of *H. vigintioctopunctata.*

## Figures and Tables

**Figure 1 biology-13-00250-f001:**
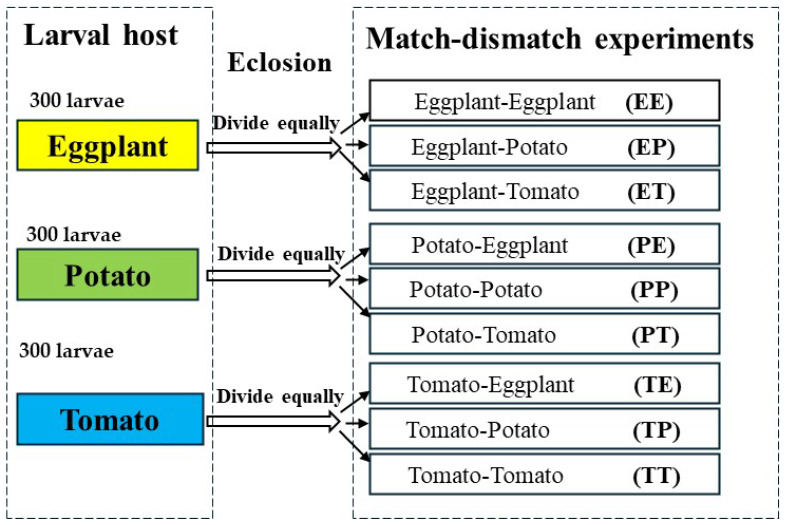
Each group, consisting of 300 *H. vigintioctopunctata* larvae, was raised on a different host plant leaf: potato, eggplant, or tomato, until all larvae reached pupation. Subsequently, we divided the emerging adults from the same host plants into three equal groups for match-mismatch experiments. *H. vigintioctopunctata* larvae were initially fed potato leaves, with one-third of the newly emerged adults subsequently fed potato leaves (Potato-Potato, PP), while the remaining two groups were fed egg-plant leaves (Potato-Eggplant, PE) and tomato leaves (Potato-Tomato, PT), respectively. Similarly, *H. vigintioctopunctata* larvae were initially fed on eggplant leaves, and then one-third of the newly emerged adults were fed eggplant (Eggplant-Eggplant, EE), potato (Eggplant-Potato, EP), and tomato (Eggplant-Tomato, ET) leaves, respectively. In a similar vein, larvae fed on tomato leaves, and then one-third of the newly emerged adults were fed tomato (Tomato-Tomato, TT), potato (Tomato-Potato, TP), and eggplant (Tomato-Eggplant, TE) leaves, respectively.

**Figure 2 biology-13-00250-f002:**
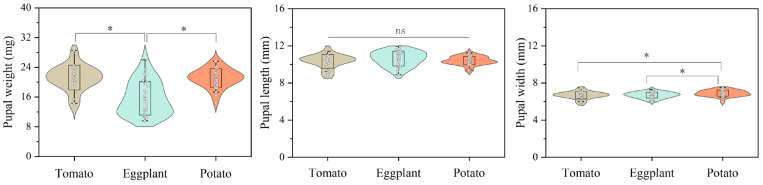
The effects of host plants on pupal weight, length and width of *H. vigintioctopunctata*. “ns” indicates no significant difference in the pupal length of *H. vigintioctopunctata* when fed on tomato, eggplant, and potato. “*” means significant difference in the pupal weight and pupal width of *H. vigintioctopunctata* when fed on tomato, eggplant, and potato.

**Figure 3 biology-13-00250-f003:**
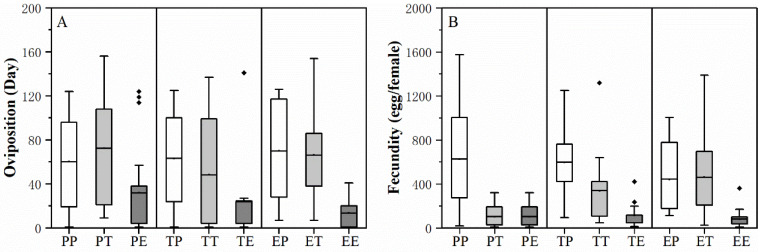
The effect of larval and adult host plant experiences on the oviposition period (**A**) and fecundity (**B**) of *H. vigintioctopunctata* adults. Post-hoc analyses based on the GLMM were used to compare significant differences in the oviposition period and fecundity of *H. vigintioctopunctata* adults across different treatments. *H. vigintioctopunctata* were reared either continuously as larvae and adults on potato (PP), tomato (TT), or eggplant (EE), or as larvae on potato and then as adults on tomato (PT) and eggplant (PE), as larvae on tomato and then as adults on potato (TP) and eggplant (TE), or as larvae on eggplant and then as adults on potato (EP) and tomato (ET). The box plots display the mean with whiskers extending from the minimum to the maximum value.

**Figure 4 biology-13-00250-f004:**
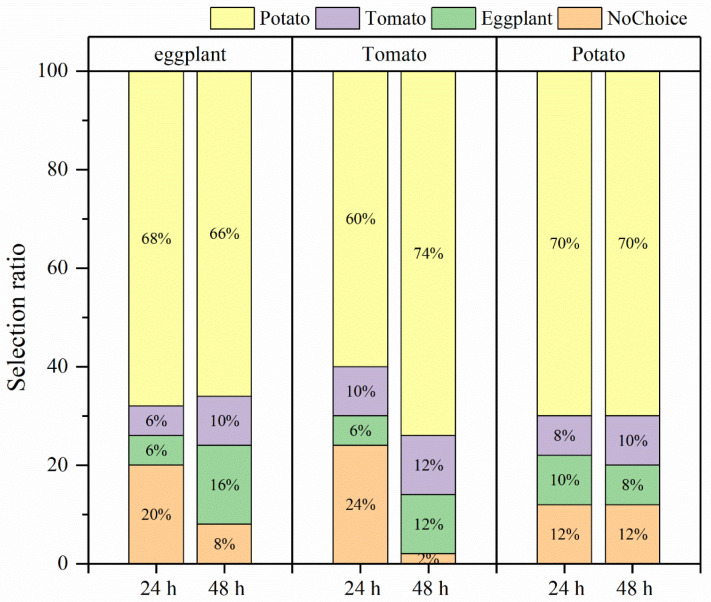
The selection ratio of *H. vigintioctopunctata* adults with different larval host experience. The upper x-axis represents the larval host feeding experience. Yellow, purple, green, and orange indicate the selection ratio for potato, tomato, eggplant, and nochoice within 24 h and 48 h, respectively. NoChoice indicates that the *H. vigintioctopunctata* adults were not found on the host plants but elsewhere.

**Figure 5 biology-13-00250-f005:**
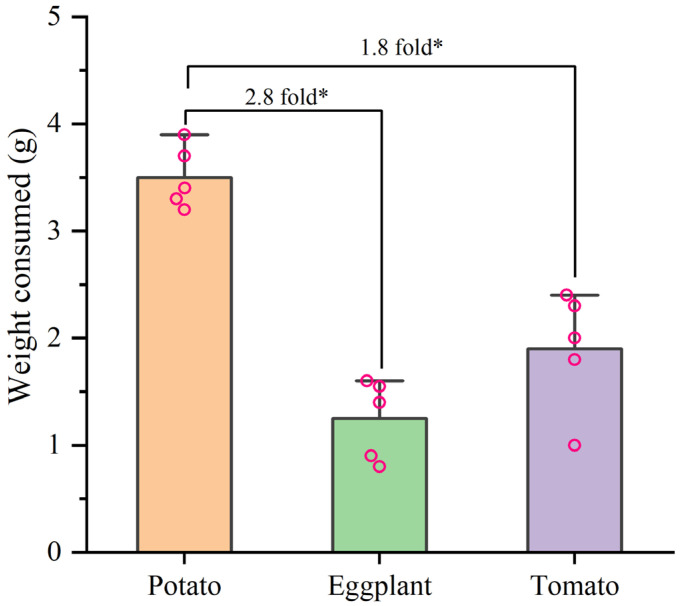
The weight consumed by *H. vigintioctopunctata* adult on different host plants in the preference experiment. * Indicates a significant difference at 0.05 level.

**Figure 6 biology-13-00250-f006:**
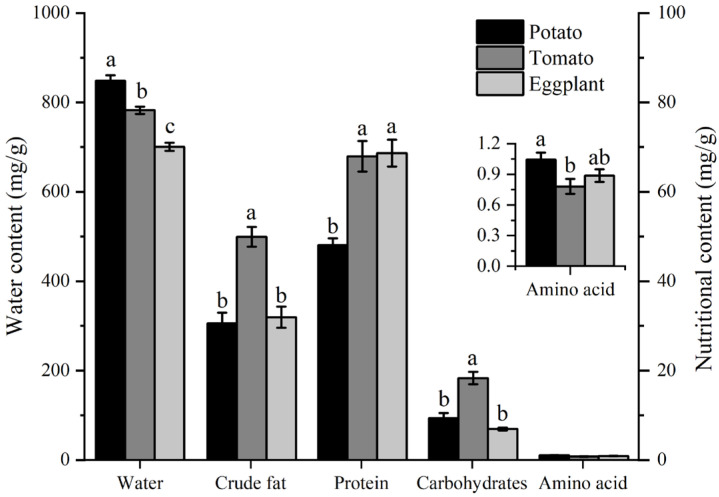
The contents of water and nutrient components in the leaves of different host plants of *H. vigintioctopunctata*. The different lowercase letters indicate significant differences in water and nutritional content among potato, tomato, and eggplant. The presence of different lowercase letters on water and nutritional content indicates of potato significant differences. The y-axis on the left illustrates the water content across various host plants, whereas the right y-axis depicts the levels of crude fat, protein, carbohydrates, and total amino acids in these plants.

**Table 1 biology-13-00250-t001:** Mean (±SE) of developmental time (d) and survival rate (%) of *H. vigintioctopunctata* reared on potato, tomato and eggplant.

Stage	Host Plant		
	Potato	Tomato	Eggplant
L1 (d)	3.47 ± 0.09 a	3.65 ± 0.09 a	3.73 ± 0.09 a
L2 (d)	2.88 ± 0.09 b	2.62 ± 0.06 c	3.17 ± 0.10 a
L3 (d)	2.85 ± 0.10 b	2.82 ± 0.07 b	3.75 ± 0.08 a
L4 (d)	5.18 ± 0.13 c	5.63 ± 0.06 b	6.00 ± 0.09 a
Pupa (d)	3.78 ± 0.06 a	3.73 ± 0.05 a	3.72 ± 0.05 a
Total larval stage (d)	14.28 ± 0.19 a	15.11 ± 0.09 b	16.61 ± 0.19 a
Larval survival rate (%)	68.0 ± 5.5 b	91.0 ± 3.4 a	74.9 ± 7.7 ab

The means followed by different letters in the same row are significantly different between different host plant at 5% significance level. L1 (the first instar); L2 (the second instar); L3 (the third instar); L4 (the fourth instar) and d (days).

**Table 2 biology-13-00250-t002:** Influence of the larval and adult host experience as well as their interaction on life history performance of *H. vigintioctopunctata* adult.

Stage	Factor	*F* Value	*p* Value
Preoviposition period (d)	Larval host	0.600	0.549
	Adult host	1.073	0.344
	Larval * adult host	0.883	0.475
Oviposition days (d)	Larval host	0.467	0.627
	Adult host	15.839	<0.0001 *
	Larval * adult host	0.786	0.536
Female adult (d)	Larval host	2.585	0.0780
	Adult host	2.457	0.0883
	Larval * adult host	1.229	0.299
Male adult (d)	Larval host	0.251	0.778
	Adult host	0.575	0.564
	Larval * adult host	1.722	0.148
Fecundity (eggs)	Larval host	1.410	0.322
	Adult host	37.289	<0.0001 *
	Larval * adult host	5.557	<0.0001 *

The *F*-value and *p*-value were calculated using a generalized linear mixed-effects model (GLMM). An asterisk (*) indicates a significant difference at the 0.05 level.

## Data Availability

Data are contained within the article.

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
