# Peer review of "Adult Feeding Experience Determines the Fecundity and Preference of the Henosepilachna vigintioctopunctata (F.) (Coleoptera: Coccinellidae)"

_biology, 2024, doi:10.3390/biology13040250_

Round 1

Reviewer 1 Report

Comments and Suggestions for Authors

H. vigintioctopunctata is well known herbivorous beetle, known to feed on various plants from the family Solanacea. Its feeding on these plants and performace as pest is well documented. I am not convinced that the study here presented would ever give other results. These beetles are known to feed on all three studied plants in a broad geographical range. Moreover, H. vigintioctopunctata is a very common species (or group of cryptic species) widely distributed in almost whole Asia. Thus there were rather no doubts that is non-selective species. As all these plants are introduced from Aouth America, and H. vigintioctopunctata is a native Asian species, it would rather be more interesting to check all these feeding and developmental data comparing native Asian solanaceaeand cultivated potatoes, egg plants and tomatoes. Below several more technical comments:

- all sentences should start with a word not number, check text throughout
- certainly there is something wrong with weights and sizes (heading 3.1), as these beetles never exceed 1 cm long as adults or larvae and certainly they do not weight 15 g..
- x and y axis on the figure 1 must be described in English not Chinese
- in all tables 'd' symbol must be explained
- heading 3.3 - Latin species names need to be italicised

Reviewer 2 Report

Comments and Suggestions for Authors

The submitted manuscript ‘Adult feeding experience determines the fecundity and prefer-2 ence of the Henosepilachna vigintioctopunctata’ by Jingwei Qi, Xiangping Wang, Tingjia Zhang, Chuanren Li and Zailing Wang contains a quite interesting material which gives an insight into the biology and ecology of the noxious coccinellid betle. The material is interesting and of practical value. Nevertheless, the manuscript needs some corrections and supplementing with data and clear descriptions. Below, there are major issues that need improvements.

Please, provide the taxonomic information in the abstract if not in the title (i.e., the author, and systematic position). It appears only in the introduction. In the Simple summary, it would be advisable to include also the common name of the ladybeetle.

l. 53: please, provide the taxonomic data of the species

l.  60-62: please, correct the sentence order

l.116-121: please, italicize the Latin names and do the same in the whole manuscript

Paragraph 2.3

The description is very complicated and unclear. My questions arise probably from my misunderstanding of the design.

134-135: I guess, it should have been ‘or’ and not ‘and’ as you did not switch the food among the experimental groups

136: I suppose that ‘a new leaf’ of the same species of the host plant was given to the larvae and not from the ‘different’ host plant. This was probably the control group. Please, clarify. I would suggest a more clear description, for example: ‘I group (300 larvae) potato – potato – control? Group II (300 larvae) potato –eggplant; Group III (300 larvae) potato-tomato’. How long did the experiment last? You put 10 larvae in one Petri dish, so you actually made 30 replications of each experiment, as I understand? Or there were actually 9 groups – three controls – match experiments (potato, tomato, eggplant) and 6 mismatch, how many insects each? – And how many replications? Probably 10 replications and 10 larvae in each replication. Please, provide these details. Also, provide these details in the descriptions to the tables and figures. Also, the descriptions should contain the name of the statistical test and the p value. The number of replications should be given together with the number of insects per replicate as well.

How many adults emerging from pupae were finally used in the mismatch experiment? The data for the survivorship, male and female longevities were collected in a separate experiment, I presume. Otherwise, you would have to wait until all adults died. Please, clarify the process of collecting data and provide more accurate details of the experiments.

Paragraph 2.4

How many insects per replicate? Was it 100x5 or 20x5? How were the plants arranged in the chamber? How many plants were there? Where were the beetles released? Were there whole plants or plant leaves? Were the plant leaves measured before the experiment? Please provide more details of the experiment.

Results

Figure 1: Please provide the descriptions in the figure in English

Figure 3: What is no choice? The plant leaves that were not eaten by the beetles? Or beetles recorded not on the plants but elsewhere? Please, explain.

l. 250-253: Please, provide more accurate data – means and SD and statistical analysis. A separate table would be useful for this analysis.

Discussion

Paragraph 4.1

You discuss only the eggplant secondary chemistry. What about potato and tomato? All of them are Solanaceae whose bodies are full of alkaloids and other allelochemicals. The discussion should be more broad in respect of plant allelochemicals.

Also, in discussing the nutritional values, you do not provide any real data to support your hypothesis. Are you going to publish the results mentioned in the lines 323-328? If possible, please provide the journal where you are going to submit the material.

l. 382-385: there was no correlation analysis performed or you did not include the results. Please, clarify.

Round 2

Reviewer 2 Report

Comments and Suggestions for Authors

I am satisfied with the changes and additions provided by the Authors.

Comments on the Quality of English Language

Some editing is required concerning the use of tenses and word order.